Comparison of histomorphology and DNA preservation produced by fixatives in the veterinary diagnostic laboratory setting

Craft William F. 1
Conway Julia A. 1
Dark Michael J. 1 2 darkmich@ufl.edu
1 Department of Infectious Diseases and Pathology, College of Veterinary Medicine, University of Florida , Gainesville, FL , USA
2 Emerging Pathogens Institute, University of Florida , Gainesville, FL , USA
Kass Philip
Electronic publication date: 2014 May 6
Publication date: 2014
Volume: 2
Electronic Location ID: e377
Received 2014 Mar 20; Accepted 2014 Apr 22
Copyright: © 2014 Craft et al.
Copyright year: 2014
Copyright holder: Craft et al.
License: This is an open access article distributed under the terms of the Creative Commons Attribution License, which permits unrestricted use, distribution, reproduction and adaptation in any medium and for any purpose provided that it is properly attributed. For attribution, the original author(s), title, publication source (PeerJ) and either DOI or URL of the article must be cited.
License URL: https://creativecommons.org/licenses/by/4.0/

Keywords: Fixative, PCR, Formalin, Histomorphology, Veterinary

Funding: University of Florida Emerging Pathogen Institute Funding was provided through University of Florida Emerging Pathogen Institute startup funds to MJD. The funders had no role in study design, data collection and analysis, decision to publish, or preparation of the manuscript.

==============================
Histopathology is the most useful tool for diagnosis of a number of diseases, especially cancer. To be effective, histopathology requires that tissues be fixed prior to processing. Formalin is currently the most common histologic fixative, offering many advantages: it is cheap, readily available, and pathologists are routinely trained to examine tissues fixed in formalin. However, formalin fixation substantially degrades tissue DNA, hindering subsequent use in diagnostics and research. We therefore evaluated three alternative fixatives, TissueTek® Xpress® Molecular Fixative, modified methacarn, and PAXgene®, all of which have been proposed as formalin alternatives, to determine their suitability for routine use in a veterinary diagnostic laboratory. This was accomplished by examining the histomorphology of sections produced from fixed tissues as well as the ability to amplify fragments from extracted DNA. Tissues were sampled from two dogs and four cats, fixed for 24–48 h, and processed routinely. While all fixatives produced acceptable histomorphology, formalin had significantly better morphologic characteristics than the other three fixatives. Alternative fixatives generally had better DNA amplification than formalin, although results varied somewhat depending on the tissue examined. While no fixative is yet ready to replace formalin, the alternative fixatives examined may be useful as adjuncts to formalin in diagnostic practices.

Introduction

Histopathology is the most useful tool for diagnosis of a number of diseases, especially cancer. To be effective, histopathology requires that tissues be fixed prior to processing. The ultimate tissue fixative for histopathology would create tissue histomorphology identical to formalin, pose no hazard to human health, preserve nucleic material for an extended period of time preferably at room temperatures, and be cost effective. Formalin has been the most common histologic fixative for over 100 years (Fox et al., 1985) since it replaced alcohol fixation. Formalin fixation offers many advantages: primarily, it is cheap and readily available. Pathologists are routinely trained to examine formalin-fixed tissues (Gugic et al., 2007; Srinivasan, Sedmak & Jewell, 2002) and currently rarely see alcohol-fixed tissues. Many tissue qualities relied upon by pathologists to make diagnoses, such as tinctorial qualities and texture, are dependent on fixation methods (Fox et al., 1985).

However, formalin has several disadvantages as well. It can cause respiratory irritation and is classified as a carcinogen (Bolt, Degen & Hengstler, 2010; Bosetti et al., 2008; Buesa, 2008; Gugic et al., 2007). Formalin also degrades DNA, RNA, and proteins, which makes formalin-fixed tissues less usable for downstream molecular diagnostics (Buesa, 2008; Gugic et al., 2007). Formalin is an aldehyde-based fixative that works by cross-linking proteins, which irreversibly damages proteins and nucleic acids (Srinivasan, Sedmak & Jewell, 2002). Typically, nucleic acids degrade in a time- and length-dependent manner (Nam et al., 2014). Protein degradation is dependent on the protein selected, its intracellular location, and the downstream application (Kothmaier et al., 2011).

The most common method for preserving tissues for molecular diagnostics is freezing at −80 °C, although this does not allow for histopathologic examination of tissues. This method itself has a number of issues, including requiring special equipment, difficulty in shipping samples to laboratories, and requiring duplicate samples to be taken for histopathologic examination.

This is impractical for many private veterinary practices, as these generally lack the facilities to freeze samples at −80 °C and many samples are too small to duplicate samples. Therefore, alternative fixatives have been proposed to allow for both histopathologic examination and molecular diagnostics (Cox et al., 2006; Gugic et al., 2007; Kap et al., 2011; Vincek et al., 2003). These have been shown to preserve nucleic acids with results similar to those obtained with fresh or frozen tissues, while still preserving histomorphology. Some of the more successful alternatives include Tissue-Tek® Xpress® Molecular Fixative (Gugic et al., 2007), PAXgene® (Kap et al., 2011), and modified methacarn solution (Cox et al., 2006). All of these are alcohol-based and non-cross-linking. Evaluation of histomorphology preservation has varied among the studies evaluating these fixatives and generally involve research settings using techniques that are not practical in most clinical situations. In addition, some of these fixatives are currently cost prohibitive in the veterinary clinical setting. The majority of the studies evaluating these fixatives have evaluated single organs from humans or rodents (Cox et al., 2006) or multiple organs from humans (Kap et al., 2011; Vincek et al., 2003). One study evaluated Tissue-Tek® Xpress® Molecular Fixative and formalin comparing histomorphology and RNA quality from a variety of animal tissues (small animals, rodents, lagamorphs, birds, insects, and lizards) both at room temperature and high ambient temperatures simulating field collection of samples (Gugic et al., 2007). They concluded that Tissue-Tek® Xpress® Molecular Fixative protected RNA and provided acceptable histomorphology that would not hinder histologic diagnosis in the species studied. Some studies evaluating multiple animal species have included limited numbers of fixatives for comparison (Gugic et al., 2007; Vincek et al., 2003).

The main limitation of all of these previous studies is that they have evaluated fixatives in a research setting. There has not been a systematic evaluation of these to determine their utility in the veterinary diagnostic setting. Alternative fixatives would have a number of benefits for veterinary diagnostic laboratories, including (depending on the nature of the fixative) decreasing hazardous waste disposal costs, decreasing health risks to laboratory workers, and enhancing the power of retrospective studies. Therefore, we conducted this study to determine how alternative fixatives would function in a standard diagnostic laboratory setting by evaluating histomorphology of a variety of tissues from dogs and cats, as well performing a quantitative evaluation of recoverable DNA from tissues.

Materials and Methods

All study protocols were approved by the University of Florida Institutional Animal Use and Care Committee (approval #201105654), and all animals were euthanized for reasons unrelated to this project. Necropsies were performed on four cats that had finished another research study and two shelter dogs within four hours of euthanasia. Replicate samples, approximately 10 × 10 × 5 mm, were collected from the liver, brain, lung, mesenteric lymph nodes, kidney, and spleen. One sample of each tissue was frozen at −80 °C. The remaining samples were placed into 10% neutral buffered formalin, Tissue-Tek® Xpress® Molecular Fixative (TT-XMF), modified methacarn, and PAXgene®, with a minimum of 1:10 tissue to fixative volume. Samples were allowed to fix for 24–48 h at room temperature with the exception of tissues in PAXgene®, which were fixed and preserved according to the manufacturer’s protocol. In brief, tissues were trimmed, placed into the supplied cassettes, immersed into chamber one for 4 h, and then switched into chamber two for 24–48 h prior to processing.

All tissues were processed with a Tissue-Tek processor using a standard overnight protocol (excluding formalin steps) followed by paraffin embedding and hematoxylin and eosin staining. The 10% neutral buffered formalin, Tissue-Tek® Xpress® Molecular Fixative, and PAXgene® fixatives were purchased commercially (ThermoFisher Scientific, Waltham, MA). Modified methacarn was prepared as previously described, using 8 parts methanol and 1 part glacial acetic acid (Cox et al., 2006).

Histomorphology was evaluated by two blinded board-certified veterinary anatomic pathologists (MJD, JAC) and one blinded anatomic pathology resident (WFC). Histomorphology of nuclear, cytoplasmic, and cellular membrane detail were evaluated on a 1–4 scale (Table 1). Sample scores were averaged between all three evaluators. For one cat, the formalin-fixed lymph node sample was lost from the block; therefore, formalin fixation histomorphometry scores for lymph node are based on the remaining five samples. Both the individual components of the histomorphometry score as well as the total score were evaluated using a Kruskal–Wallis test (Lowry, 2012) to determine if there was a difference between any of the four groups. If a significant difference was found (p < 0.05), the Mann–Whitney test was used to compare each group to each other group, to determine significant differences between each individual fixative. P-values were then adjusted using the Holm-Bonferroni correction for multiple comparisons, using the p.adjust function of R v3.0.2. Fixatives were considered significantly different if the Holm-Bonferroni corrected two-tailed Mann–Whitney p value was less than 0.05. The minimum, 25th quartile, median, 75th quartile, and maximum were calculated for each tissue as well as for all tissues combined using Microsoft Excel (v14.3.9; Microsoft Corp., Seattle, WA). Graphs were generated using GNUplot (v.4.6, patchlevel 3).

Table 1 Histomorphology scoring criteria.

Characteristic	Score	Criteria	
Nuclear	4	Sharp nuclear membrane; chromatin pattern clear; nucleolus, when present is distinct	
	3	Slight degradation in chromatin pattern, nucleolus when present, less distinct but discernable, sharp nuclear membrane	
	2	Less distinct nuclear membrane; fuzzy chromatin pattern, nucleolus when present is difficult to discern	
	1	Fuzzy nuclear membrane, chromatin pattern difficult to determine, nucleoli indetectable	
	0	Nucleus not able to be differentiated from cytoplasm	
Cytoplasm	4	Normal cellular morphology easily determined	
	3	Intracytoplasmic details fuzzy	
	2	Only rare evidence of normal intracellular structures	
	1	Increased cytoplasmic pallor, increased cytoplasmic eosinophilia	
	0	Cytoplasm homogenously pale eosinophilic with no evidence of organelles	
Cell Membranes	4	Cells have distinct intracellular; any normal substructures, if present, are easily distinguished	
	3	Loss of substructures (if present) in some cells; slight loss of intracellular details	
	2	Loss of substructures (if present) in most cells; obvious blurring of many cellular borders	
	1	No substructures detected; significant blurring of most cellular borders	
	0	Cells unable to be distinguished from adjacent cells	

Tissue scrolls were obtained from the paraffin blocks one week after processing and DNA was extracted using the QIAamp DNA FFPE Tissue kit (Qiagen Inc., Valencia, CA). Primers were designed by aligning the sequences of the retinol-binding protein 3, interstitial gene (IRBP) from dog, mouse, rat, and human, and selecting regions that were relatively conserved, to generate 100, 200, 300, 500, and 750 base pair long amplicons (Table 2). Each reverse primer was combined with the forward primer and combined with extracted DNA. These reactions were amplified via PCR on an Applied Biosystems Veriti Thermal Cycler with the following conditions: 96 °C for 3 min, followed by 35 cycles of 96 °C for 1 min, 60 °C for 1 min, then 72 °C for 1 min. This was followed by 7 min at 72 °C, with a final hold at 4 °C until the next morning. Samples were examined on a 1.25% agarose gel via electrophoresis to ensure a single band of the appropriate size was generated per lane. DNA extracted from tissues frozen at −80 °C using the QIAamp DNA Mini Kit (Qiagen Inc.) and water were used as positive and negative controls, respectively.

Table 2 DNA primers used in this study.

Primer name	Sequence	
IRBP_F	CCT KGT RCT GGA NAT GGC	
IRBP_R1_100bp	CTC TTG ATG GCC TGC TC	
IRBP_R2_200bp	GGC TCA TAG GAG ATG ACC AG	
IRBP_R3_300bp	CAG GTA GCC CAC RTT NCC CTC	
IRBP_R4_400bp	CGG AGR TCY AGC ACC AAG G	
IRBP_R5_500bp	GAT CTC WGT GGT NGT GTT GG	
IRBP_R6_750bp	CTC AGC TTC TGG AGG TCC	

The presence or absence of bands for all sizes was noted. The Kruskal–Wallis test was used to determine if there was a significant difference between the maximum band size for any of the fixatives. If significant (p < 0.05), the Mann–Whitney test was used to compare each fixative against each other to determine which differences were significant. P-values were adjusted using the Holm-Bonferroni correction for multiple comparisons, using the p.adjust function of R v3.0.2; significance was determined by a corrected two-tailed p-value <0.05. The minimum, 25th quartile, median, 75th quartile, and maximum calculated for each fixative and for each tissue using Microsoft Excel. Graphs were generated using GNUplot (v.4.6, patchlevel 3).

Results

Histomorphology

While the majority of the alternative fixatives produced adequate histomorphology in the tissues examined, formalin fixed tissues consistently resulted in superior histomorphology. There was no statistically significant difference between mean histomorphology scores comparing dog and cat tissues, and these were combined for subsequent analysis. The nuclear, cytoplasmic, cellular membrane, and total scores averaged between all three examiners (Figs. 1A–1D) for formalin fixed tissues were higher than for all other fixatives (p = 0.0006), although there was substantial variation with all fixatives (Figs. 2A–2D).

Figure 1 Comparison of histomorphology and DNA preservation produced by fixatives in the veterinary diagnostic laboratory setting histomorphology scores for all animals and tissues combined.

The median is represented by a red diamond, the box represents the 25th and 75th quartiles, and the whiskers represent 1.5× interquartile range.

Figure 2 Comparison of histomorphology and DNA preservation produced by fixatives in the veterinary diagnostic laboratory setting histomorphology scores for individual tissues.

The median is represented by a red diamond, the box represents the 25th and 75th quartiles, and the whiskers represent 1.5× interquartile range.

While a number of minor artifacts were noted, the primary difference noted between formalin and the other fixatives was in erythrocytes. This is likely reflected in the significantly higher scores for formalin vs. other fixatives in the spleen (p = 0.0051), an organ made up in large part by erythrocytes.

DNA preservation

Formalin had significantly shorter total maximum DNA band sizes than modified methacarn solution (p < 0.0006) and PAXgene (p = 0.0032) (Fig. 3). In particular, the bands obtained from lymph nodes were significantly smaller with formalin than with modified methacarn (p = 0.0480) (Fig. 4). Overall, modified methacarn solution performed as well or better than the other fixatives for all tissues, with the best score in brain (median amplicon length of 750 bp).

Figure 3 Maximum DNA amplicon size ranges for all samples combined.

The median is represented by a red diamond, the box represents the 25th and 75th quartiles, and the whiskers represent 1.5 × interquartile range.

Figure 4 Maximum DNA amplicon size ranges for different tissue samples.

The median is represented by a red diamond, the box represents the 25th and 75th quartiles, and the whiskers represent 1.5 × interquartile range.

Figure 5 Representative fixative histomorphology—liver.

Samples are from the liver of a single cat. A—formalin, B—TT-XMF®, C—modified methacarn, D—PAXgene®.

Figure 6 Representative fixative histomorphology—spleen.

Samples are from the spleen of a single dog. A—formalin, B—TT-XMF®, C—modified methacarn, D—PAXgene®.

Discussion

While alternative fixatives have been found to work well in research settings (Cox et al., 2006; Kap et al., 2011; Vincek et al., 2003), these are not ready to replace formalin for routine tissue processing in the veterinary diagnostic laboratory. All of the fixatives require tissues be prevented from contacting formalin to benefit from their nucleic acid preserving qualities, which would require laboratories to either maintain separate tissue processors or bar submission of formalin-fixed tissues. Neither of these is practical in veterinary practice. Several fixatives produce excellent histomorphology with alternative processing techniques; this is also impracticable in most veterinary diagnostic laboratories, as it would require separate processing runs.

However, while no fixative is ideal from the standpoint of replacing formalin, all fixatives produced interpretable slides. Therefore, using alternative fixatives may be useful in specific circumstances where subsequent DNA isolation may be required. For example, tissue samples from neoplasms may be saved separately to generate a tissue bank for subsequent research projects. The specific alternative chosen should be based on the tissue selected, as well as predicted needs for DNA amplification and preservation of histomorphology. For example, while TT-XMF had better histomorphology scores in the kidney than either modified methacarn or PAXgene, it had a lower median DNA amplicon size.

One characteristic observed with alternative fixatives was that bloody or congested tissues often had unfixed areas, which could result in missing lesions and inaccurate diagnoses. This has not been found in previous studies (Cox et al., 2006), and may be due to a number of factors. First, the size of sample taken will greatly influence fixation. For most veterinary diagnostic laboratories, 1 cm thick samples are considered standard for histopathologic examination. In many previous studies, samples taken for fixation were substantially thinner; for example, the study by Cox et al. used 15 mm × 8 mm × 3 mm samples. Other possibilities include differences in processing; microwave fixation (Cox et al., 2006) or rapid tissue processing (Vincek et al., 2003) techniques have been used. Tuning the processing technique for the fixative selected would likely improve fixation and the ultimate histomorphology.

Finally, our evaluation of macromolecule preservation was limited to DNA. Additional analysis would be required to determine whether these fixatives preserve RNA equally well. Other variables require investigation to determine the best fixative for a particular application. These include the effects of fixation time on nucleic acid quality, as many samples will sit longer than 24 h before processing, as well as the effect of storage time after tissue processing but before sectioning for nucleic acid isolation, since many blocks will be stored for a period of time between the evaluation of histopathology and nucleic acid isolation. The latter is especially important if laboratories set up tissue banks, as samples would be expected to be stored for prolonged periods.

Conclusions

While no fixative is ideal to replace formalin, alternative fixatives have generally acceptable histomorphologic characteristics in most tissues and are valuable adjuncts to standard formalin fixation. Investigators proposing to use an alternative fixative for a research project should evaluate the project goals and requirements. Ideally, the fixative should be tested with samples of the target organs to determine the best fixative, required processing techniques, and histomorphology compromises before actual sample collection begins.

Supplemental information

Supplemental Information 1 DNA scores

Click here for additional data file.

Supplemental Information 2 Histomorphology scores

Click here for additional data file.

The authors would like to acknowledge Patrick Knisley for his assistance with samples, the University of Florida College of Veterinary Medicine Histopathology Laboratory for help with tissue processing, and Antoinette McIntosh for help with DNA processing.

Additional Information and Declarations

Competing Interests

Author Contributions

Animal Ethics

The authors declare there are no competing interests.

William F. Craft conceived and designed the experiments, performed the experiments, analyzed the data, wrote the paper, prepared figures and/or tables, reviewed drafts of the paper.

Julia A. Conway performed the experiments, analyzed the data, reviewed drafts of the paper.

Michael J. Dark conceived and designed the experiments, performed the experiments, analyzed the data, contributed reagents/materials/analysis tools, wrote the paper, prepared figures and/or tables, reviewed drafts of the paper.

The following information was supplied relating to ethical approvals (i.e., approving body and any reference numbers).

All study protocols were approved by the University of Florida Institutional Animal Use and Care Committee (approval #201105654).

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
