# Peer review of "Comparison of histomorphology and DNA preservation produced by fixatives in the veterinary diagnostic laboratory setting"

_PeerJ, doi:10.7717/peerj.377_

## Round 0.1 · original submission · Minor Revisions

In addition to addressing comments provided by the three reviewers, I am requesting additional explanation and clarification of your statistical methods. You have consistently used a one-tailed test for comparisons, but provide no justification for why the alternative hypothesis is unidirectional. A convincing explanation should be provided in lieu of your decision to not employ two-tailed tests. In addition, it is conventional to use some type of multiple comparison adjustment for post-hoc (Mann-Whitney) tests following a significant ANOVA (e.g., Kruskal-Wallis) result; this should be provided as well. Finally, on line 132 you describe your results as differences in mean scores; this is not correct. Instead, nonparametric tests such as Kruskal-Wallis and Man-Whitney contrast the distribution of scores between groups, not means or medians.

Reviewer 1 ·

Basic reporting

Not very innovative but acceptable.

Experimental design

Very simple but acceptable.

Validity of the findings

Only a minimal contribution to the progress in science but acceptable.

Additional comments

Acceptable.

Reviewer 2 ·

Basic reporting

No Comments

Experimental design

The authors compared histomorphology and DNA preservation of paraffin embedded animal tissue that was fixed using different tissue fixatives.

Validity of the findings

The authors concluded that formalin fixed tissue had superior histomorphology comparing to other alcohol based fixatives but generally alcohol based fixative had better DNA preservation.
The authors suggested based on their findings that alternative fixatives should be tested before initiating research project to determine the best fixative.
I believe that overall these conclusions are fair and that this paper will contribute to further discussion of using alternative fixative as an alternative to formalin. Therefore I recommend it to be published.
However there are several issues that could be added to discussion to make the publication better.
1. The authors did mentioned that the paper limitation is that only DNA was studied. They mentioned shortly RNA and I think would be great to add literature data about various fixatives and RNA preservation.
2. One of the reason to use alternative fixatives is also preservation of proteins. Formalin denature proteins whereas other fixative cause less damage. That would be worthwhile to discuss too
3. Comparing histomorphology of alternative fixative to formalin is great but several things have to be kept in mind. In that comparison formalin fixed tissue is set up as a gold standard for histology. Nothing can be better than gold. Even if you compare platinum to gold of course gold will be better. The same is with the formalin. If formalin is gold standard nothing can match it. You can see it in Figure 2. that none of 3 alternative fixatives are better or even the same in any analyzed category (cell membrane, cytoplasm, nucleus) as nothing by definition can be better than formalin. Pathologists, human or animal, are trained for over 80 years on formalin fixed material and what we learn as "normal" is "formalin induced artifact". All 3 alternative fixative in these paper are alcohol based fixatitive and they produce "alcohol based artifact". All of them are artifact of fixation just we are more used to formalin one.
If you go back about 100 years ago at the time when formalin was introduced as fixative there were multiple publiction, mostly in German pathology literature, decrying formalin and its histomorphology as at that time ethanol was primary fixative and everything was compared to ethanol. However due to the lower cost of formalin, and less possibility for abuse, ethanol was by 1930's stopped to be used as fixative.
Bottom line is that when we review tissue under the microscope we are trained to recognize artifact of fixation and we should not forget that they are artifact but just that we are used to them.
Therefore I would like if authors could include in their discussion
some historical perspective of tissue fixative and explain that polemic about formalin and alcohol based fixative go way back and is not so straitforward.

·

Basic reporting

This article addresses a practical question pertinent to many veterinary pathologists. The study design is sound though some additional data and a few more images would be useful. Introduction and Discussion are adequate.

Experimental design

It is difficult to envision how 4 replicate samples 1 x 1 x 0.5 cm (line 84) were collected from the listed tissues from cats and some dogs. Normal cat lymph nodes are barely of this size, and would not yield 4 such samples. The authors should indicate the conditions of the animals that were utilized and clarify what size samples were collected.

Some more detail on the PCR assay should be provided. How was it assured that only a single amplicon was present on gels? Was an identical annealing temperature used for all assays?

Validity of the findings

The issues identified above should be addressed. Also, it appears that TT-XMF did not generate any 100bp amplicons (Fig. 3). Is that correct or a feature of the 1.5x interquartile range? If the latter, the authors should consider showing the entire range since it seems difficult to rectify that 100 bp are less efficiently amplified than 750 bp.

Fig. 4 seems to indicate that neither 100 nor 200 bp products were derived from formalin-fixed tissue, which seems to contradict findings in Fig. 3?

The y axis on all figures should be scaled to show all whiskers.

Additional comments

There are some issues with reference formatting in the text (lines 53, 62, 63 and elsewhere). Fixation and processing for PaxGene should also be spelled out (Line 89) so readers can assess the practicality of this without having to pull up the manufacturer's protocol. Use of past tense is inconsistent i.e. see Line 134. Suggest to include images of the "most difficult" tissue (spleen) fixed with all fixatives.

---

## Round 0.2 · Minor Revisions

Thank you for making the requested changes. One comment of mine I believe was misunderstood, and it concerns the following statement: "If a significant difference was found (p < 0.05), the Mann-Whitney test was used to compare each group to each other group, to determine significant differences between each individual fixative. Fixatives were considered significantly different if the two-tailed Mann-Whitney p value was less than 0.05."

My comment did not pertain to whether or not you should do a post-hoc Mann-Whitney test, but rather that because you used this test for post-hoc comparisons, you should adjust your level of significance for the multiple comparisons to preserve the nominal Type I error percentage of 5% used for the Kruskal-Wallis test. If you compared each of the four groups to the others, then that requires six post-hoc tests. Please apply a Bonferroni, Bonferroni-Holm (sequentially rejective), or other multiple comparison adjustment procedure to these post-hoc tests.

---

## Round 0.3 · accepted · Accept

Thank you for making these final changes.